# Sex differences in performance and pacing in the greatest Quintuple Iron ultra-triathlon race in history: The IUTA World Championship 2024 in France

Beat Knechtle[1,2*], Luciano Bernardes Leite[3], Sasa Duric[4], Ivan Cuk[5],
Marilia S. Andrade[6], Volker Scheer[7], Pantelis T. Nikolaidis[8], Katja Weiss[2],
Thomas Rosemann[2], Pedro Forte[9,10,11]

1 Medbase St. Gallen Am Vadianplatz, St. Gallen, Switzerland, 2 Institute of Primary Care, University of Zurich, Zurich, Switzerland, 3 Department of Physical Education, Federal University of Viçosa, Viçosa, Brazil, 4 Liberal Arts Department, American University of the Middle East, Egaila, Kuwait, 5 Faculty of Sport and Physical Education, University of Belgrade, Belgrade, Serbia, 6 Department of Physiology, Federal University of Sao Paulo, Brazil, 7 Ultra Sports Science Foundation, Pierre-Benite, France, 8 University of West Attica, Athens, Greece, 9 Department of Sports, Instituto Politécnico de Bragança, Bragança, Portugal, 10 Department of Sports, Higher Institute of Educational Sciences of the Douro, Penafiel, Portugal, 11 Research Center for Active Living and Wellbeing (LiveWell), Instituto Politécnico de Bragança, Bragança, Portugal

* beat.knechtle@hispeed.ch

## Abstract

### Background

Pacing in ultra-triathlon has been investigated by analyzing lap times from Double to Deca Iron ultra-triathlon for World Cup races but not for a World Championship. The present study aimed to investigate pacing in ultra-triathletes competing in the fastest and largest World Championship in Quintuple Iron ultra-triathlon ever held in history.

### Methods

A total of 11 female and 24 male finishers who completed the 2024 Quintuple Ultra Triathlon World Championship in Colmar, France, were analyzed. Independent t-tests assessed sex-based performance variations with effect sizes (Cohen's d). A two-way ANOVA evaluated the effects of sex and performance quartiles on cycling and running, with eta squared (η²) used to measure effect sizes.

### Results

Overall, men were slower in swimming and cycling and faster in running and overall race time. The variability in lap times was similar in cycling for both women and men but higher in running for women. There was a significant interaction between sex and performance quartiles in cycling but not running. For cycling, the variability in

**Data availability statement:** All relevant data are within the paper and its Supporting Information files.

**Funding:** The author(s) received no specific funding for this work.

**Competing interests:** The authors have declared that no competing interests exist.

performance was higher in men compared to women; for running, it was similar for both women and men.

## Conclusions

The finding that women outperformed men in swimming and cycling, likely due to the elite nature of the World Championship, which featured a highly selected and committed female cohort with a high completion rate. While both sexes showed consistent pacing in cycling, women exhibited greater variability in running, possibly due to more frequent breaks.

## Introduction

Ultra-triathlon includes triathlon distances that are multiple (x) times the classic IRONMAN® distance (3.8 km swimming, 180 cm cycling, 42.2 km running). Such races are mainly held as World Cup races and are sanctioned by the IUTA (International Ultra Triathlon Association) (https://iutasport.com/iuta/about). These ultra-triathlon races include race distances of 2x (7.6 km swimming, 360 km cycling, 84.4 km running), 3x (11.4 km swimming, 540 km cycling, 126.6 km running), 5x (19 km swimming, 900 km cycling, 211 km running), 10x (38 km swimming, 1800 km cycling, 422 km running) and up to 20x (76 km swimming, 3600 km cycling, 844 km running) the IRONMAN® distance [1]. In recent years, the popularity of these races has been increasing [2].

Pacing is an important race tactic – especially in long races such as triathlon races – to finish successfully [3–5]. There are different strategies for pacing, e.g., negative, all-out, positive, even, parabolic-shaped, and variable pacing [6]. Negative pacing means an increase in speed over time, whereas positive pacing means a decrease in speed. In triathlon, many variables can be analyzed, such as the split disciplines, race length, environmental influences, and others [3,7,8].

In longer triathlons such as the IRONMAN® distance, the athletes' level also seems to influence pacing. In IRONMAN® triathletes competing in races other than the IRONMAN® World Championship, positive pacing was described in both cycling and running [9]. However, there appeared to be differences between women and men, with women reducing their cycling speed significantly more than men. However, no differences were reported for running [9]. When professional IRONMAN® triathletes competing in the IRONMAN® World Championship in Hawaii were analyzed, men showed a negative pacing in cycling, while women showed an even pace. In running, both women and men showed a positive pacing [10].

Considering ultra-triathlon, one study to date has investigated pacing in the Double Iron to the Deca Iron ultra-triathlon by analyzing changes in speed during cycling and running laps. The pacing was positive, with faster athletes showing even pacing compared to slower athletes in the Double and the Triple Iron ultra-triathlons. For longer race distances such as the Quintuple and the Deca Iron ultra-triathlon, no differences were found between faster and slower athletes [11].

If we now look at the differences between IRONMAN® races and the IRONMAN® World Championships, no study has analyzed a World Championship for ultra-triathletes. Therefore, in the present study, we analyzed the largest and fastest World Championship in the Quintuple Iron ultra-triathlon, which took place in Colmar, France, in 2024 (https://bretzelultratri.com/). In this race, 11 women and 24 men officially crossed the finish line. The women finished second and third overall, with the first woman setting a new world record and the second woman finishing faster than the old world record. Based on reports of IRONMAN® triathletes of different levels, we hypothesized that there would be differences in pacing at a World Championship.

## Method

### Ethical approval

This study was approved by the Institutional Review Board of Kanton St. Gallen, Switzerland, with a waiver of the requirement for informed consent of the participants as the study involved the analysis of publicly available data (EKSG 01/06/2010). It was conducted in accordance with the recognized ethical standards of the Declaration of Helsinki, adopted in 1964 and revised in 2013.

### The race

The Quintuple Ultra Triathlon IUTA World Championship was held in Colmar, France, from 24th to 30th June 2024. A total of 11 women and 33 men competed, with all 11 women finishing the race but 9 men dropping out, leaving only 24 men to officially finish the race. Notably, the two fastest women were faster than the second fastest man, making one man and two women the three fastest in the race. The male participants in the Quintuple Iron ultra-triathlon spanned a competitive age range from 30 to 54 years, as reflected by their age categories: M30 (30–34 years), M35 (35–39 years), M40 (40–44 years), M45 (45–49 years), and M50 (50–54 years). The largest male representation was in the M50 category with 7 athletes, followed by M40 (5), M45 (4), M35 (2), and M30 (1). The female participants spanned an age range from 30 to 64 years, with categories including W30, W40, W45, W50, and W60. The most common category among women was W40 (6 participants), followed by W50 (2), and one participant each in W30, W45, and W60. This distribution highlights a predominance of women aged 40–44 years, but also reflects a broader age diversity among the female competitors.

### Data set and data preparation

The race data with split and lap times for swimming, cycling, and running were obtained from the official race website of the BRETZEL ULTRA TRI (https://bretzelultratri.com/quintuple-continu-day/). All swimming, cycling, and running courses are presented on the race website (https://bretzelultratri.com/en/race-course/). The swimming, which covered 19 km with 380 laps of 50 m, took place in the Stade Nautique outdoor pool in the city of Colmar (www.colmar.fr/stade-nautique). After swimming, the athletes had to cover a transfer route of 11.25 km to the cycling course. During this transfer, each athlete was escorted by two cyclists for safety reasons. The cycling course had 103 laps of 8.752 km in the Waldeslust region near Colmar to achieve 900 km. The cycling was carried out as a non-drafting time trial race on a relatively flat circuit, with the athletes being supported by their crew. After cycling, the athletes had to complete 159 running laps of 1.33 km. The run course was mainly on asphalt, with a small section in the forest where the course was rather uneven on large stones. The time limits were 14 hours for swimming, 81 hours for swimming and cycling, and 148 for the full distance. It should be noted that the lap times for swimming were recorded manually, while the lap times for cycling and running were recorded electronically using a chip system (www.raceresult.com/). Times are presented in seconds (s) as s is the international unit for time (https://www.phyley.com/second).

### Statistical analysis

All statistical analyses were performed using Python™ (SciPy and Statsmodels), with descriptive statistics calculated for each triathlon discipline and stratified by sex. Boxplots were used to visualize differences, and independent t-tests

assessed sex-specific performance variations, with effect sizes (Cohen's d) categorized as small, medium, or large (0.2, 0.5, and 0.8, respectively). A two-way ANOVA evaluated the effects of sex and performance quartiles on cycling and running, with eta squared ($\eta^2$) used to measure effect sizes. Transition times (T1 and T2) were analyzed separately using t-tests. A significance level of $p < 0.05$ was maintained, and Bonferroni correction was applied for multiple comparisons. Results were presented with 95% confidence intervals, and visualizations were created using Seaborn and Matplotlib.

## Results

Regarding overall ranking, women came in second and third place. The first woman set a new world record, and the second woman was faster than the old world record (Table 1).

### Performance metrics in the triathlon disciplines by sex

Table 2 presents the descriptive statistics of times in the different triathlon disciplines (cycling, running, swimming, transitions, and total time) by sex. Men had a higher mean time in cycling (44,313.04 s vs. 37,277.36 s) and swimming (28,257.50 s vs. 26,770.91 s), while women exhibited a higher mean time in running (49,789.00 s vs. 45,082.33 s). Transitions 1 and 2 had similar mean times, with transition 2 showing greater variability in women. Regarding total time, men had a lower mean (37,5665.67 s) and greater variability, highlighting the differences in the distribution of times across sexes and disciplines. This could be explained by the number of participants who completed the race (11 women and 24 men).

### Sex-based comparisons of triathlon performance metrics

Fig 1 presents the distribution of times across different triathlon disciplines (swimming, cycling, running, and total time), stratified by sex. Statistical analyses revealed no statistically significant differences between the sexes in any of the variables analyzed (p > 0.05 | Swim time: t = 0.980, p = 0.336; T1 time: t = 0.752, p = 0.463; Bike time: t = 0.213, p = 0.834; T2 time: t = −1.019, p = 0.330; Run time: t = −0.819, p = 0.427). However, the effect sizes provide insights into the magnitude of these differences. Swimming (effect size: −0.33) and cycling (effect size: −0.08) showed small effect sizes, indicating minimal practical differences. Running (effect size: 0.32) also exhibited a small effect, suggesting that men tend to be slightly faster. The effect size was small for the total time (effect size: 0.14), indicating no substantial difference between sexes. To summarize, although differences in times between men and women exist, they are small and not statistically significant.

Fig 2 shows the distribution of times for transition 1 (T1) and transition 2 (T2), categorized by sex. Statistical analyses showed no significant differences between sexes in T1 and T2 times. However, the effect size for T1 (−0.28) was moderate, indicating a potential trend of practical significance, whereas the effect size for T2 (0.02) was small, suggesting minimal practical differences.

### Lap time performance metrics by sex

Table 3 displays the descriptive statistics for mean lap times in cycling and running, stratified by sex. On average, women recorded slightly faster cycling laps (962.69 s) compared to men (985.60 s). In contrast, the opposite trend was observed in running, where men exhibited lower mean lap times (740.90 s) than women (771.75 s). Variability in lap performance, indicated by standard deviation, was slightly higher for men in both disciplines. This suggests greater dispersion in pacing strategies or physiological capacities among male participants.

The minimum and maximum values reinforce this heterogeneity. Men's cycling lap times ranged from 355.83 s to 1151.29 s, while women's ranged from 628.50 s to 1130.32 s. Similarly, in running, men's laps ranged from 612.00 s to 861.97 s, and women's from 658.04 s to 895.36 s. These results point to a broader spread among male athletes, particularly in cycling, possibly due to differences in training background, equipment, or pacing tactics. Nonetheless, it is important to consider the unequal sample sizes, 24 men and 11 women, which may influence the distribution patterns.

**Table 1. Overall ranking of all male and female participants and final time (h:min:s).**

| Classification | Swim | Bike | Run | Total |
|---|---|---|---|---|
| 1st Male | 05:49:00 | 34:20:40 | 27:43:17 | 68:35:42 |
| 1st Female | 06:10:00 | 33:39:26 | 33:08:38 | 73:52:09 |
| 2nd Female | 06:23:00 | 38:22:26 | 33:54:06 | 81:19:53 |
| 2nd Male | 06:23:00 | 35:10:31 | 43:21:59 | 85:29:55 |
| 3rd Male | 07:04:00 | 40:30:20 | 39:31:17 | 87:30:50 |
| 4th Male | 06:57:00 | 47:27:54 | 33:55:24 | 89:09:00 |
| 5th Male | 06:13:00 | 45:56:14 | 38:44:39 | 92:04:09 |
| 3rd Female | 07:21:00 | 48:05:40 | 36:32:38 | 92:22:49 |
| 6th Male | 06:37:00 | 41:53:22 | 46:38:14 | 95:31:03 |
| 7th Male | 07:36:00 | 48:45:52 | 40:15:26 | 96:58:23 |
| 4th Female | 06:41:00 | 44:46:46 | 46:31:03 | 98:39:30 |
| 8th Male | 08:46:00 | 49:57:10 | 39:46:40 | 98:45:08 |
| 9th Male | 09:52:00 | 48:18:36 | 40:11:20 | 99:02:00 |
| 10th Male | 08:11:00 | 46:10:03 | 45:56:44 | 100:28:55 |
| 11th Male | 07:14:00 | 46:47:07 | 45:56:43 | 101:27:34 |
| 12th Male | 06:52:00 | 47:29:58 | 46:01:08 | 101:33:04 |
| 13th Male | 06:30:00 | 48:46:06 | 46:54:00 | 102:49:19 |
| 14th Male | 06:42:00 | 40:26:39 | 55:54:24 | 103:20:28 |
| 15th Male | 09:34:00 | 51:34:58 | 41:37:39 | 104:17:23 |
| 5th Female | 07:36:00 | 52:51:00 | 40:46:05 | 104:40:56 |
| 6th Female | 07:34:00 | 51:28:18 | 47:19:54 | 106:55:36 |
| 16th Male | 08:31:00 | 50:55:37 | 48:37:58 | 108:33:26 |
| 17th Male | 06:25:00 | 51:52:16 | 52:07:14 | 110:44:06 |
| 18th Male | 07:54:00 | 50:48:12 | 53:35:08 | 112:50:07 |
| 7th Female | 07:55:00 | 54:20:07 | 50:22:32 | 113:01:25 |
| 19th Male | 10:20:00 | 51:55:09 | 53:09:52 | 115:53:44 |
| 20th Male | 10:15:00 | 63:23:34 | 45:59:06 | 120:08:34 |
| 8th Female | 07:45:00 | 45:39:35 | 68:42:14 | 122:24:04 |
| 21st Male | 06:44:00 | 65:09:46 | 50:03:36 | 122:32:19 |
| 21nd Male | 08:38:00 | 66:40:06 | 48:44:06 | 124:19:08 |
| 23rd Male | 09:46:00 | 64:45:57 | 49:42:50 | 124:58:51 |
| 9th Female | 07:59:00 | 61:21:49 | 55:15:56 | 125:23:20 |
| 10th Female | 06:37:00 | 56:47:22 | 62:35:25 | 126:24:35 |
| 11th Female | 09:47:00 | 58:31:42 | 60:59:28 | 129:32:39 |
| 24th Male | 09:30:00 | 68:19:06 | 58:04:12 | 137:23:08 |

## Influence of sex and performance quartiles on cycling and running times

Table 4 contains data on the effects of sex and quartiles on cycling and running performance. The cycling quartiles ($F = 35.80$, $p < 0.001$, $\eta^2 = 0.38$) and running quartiles ($F = 35.90$, $p < 0.001$, $\eta^2 = 0.38$) explained a significant portion of the variability in times. Additionally, a significant interaction between sex and cycling quartiles was observed ($F = 8.87$, $p < 0.001$, $\eta^2 = 0.095$), indicating differences in the relationship between performance quartiles and sex. For running, this interaction was also significant ($F = 5.76$, $p = 0.005$, $\eta^2 = 0.062$), suggesting some influence of sex across performance quartiles.

**Table 2. Minimum, maximum, and average performance metrics in the triathlon disciplines by sex. Times are expressed in seconds (SI).**

| Sex | Disciplines | Minimum (s) | Maximum (s) | Mean (s) | SD |
|---|---|---|---|---|---|
| Women | Cycling | 121,166 | 220,909 | 178,659.18 | 30,705.57 |
| | Running | 119,318 | 247,334 | 175,461.72 | 43,623.46 |
| | Swimming | 22,200 | 35,220 | 26,770.91 | 3,626.48 |
| | Transition 1 | 478 | 1,767 | 975.45 | 384.07 |
| | Transition 2 | 10 | 11,251 | 2,551.54 | 3,838.74 |
| | Total | 265,929 | 464,580 | 384,257.91 | 67,532.80 |
| Men | Cycling | 123,640 | 245,946 | 181,113.04 | 33,752.79 |
| | Running | 99,797 | 209,052 | 163,882.33 | 25,308.19 |
| | Swimming | 20,940 | 37,200 | 28,257.50 | 5,151.85 |
| | Transition 1 | 574 | 1,722 | 1,074.75 | 310.84 |
| | Transition 2 | 34 | 4,286 | 1,337.33 | 1,397.01 |
| | Total | 246,942 | 494,588 | 375,665.67 | 54,572.92 |
| Total | Cycling | 121166 | 245,946 | 18,0341.83 | 32,393.04 |
| | Running | 99,797 | 247,334 | 16,7521.57 | 31,980.25 |
| | Swimming | 20,940 | 37,200 | 27,790.29 | 4,723.65 |
| | Transition 1 | 478 | 1,767 | 1,043.54 | 333.07 |
| | Transition 2 | 10 | 11,251 | 1,718.94 | 2,445.69 |
| | Total | 246,942 | 494,588 | 378,366.09 | 58,072.56 |

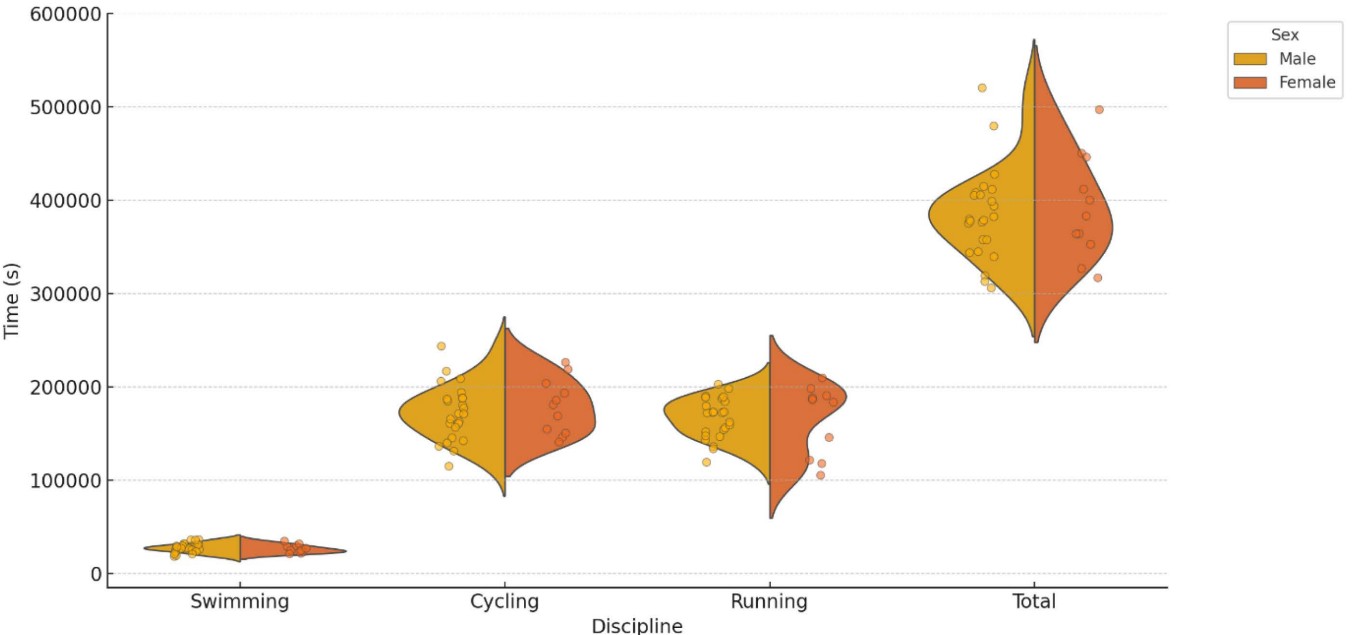

**Fig 1. Distribution of times by sex for Cycling, Running, Swimming, and Total times.**

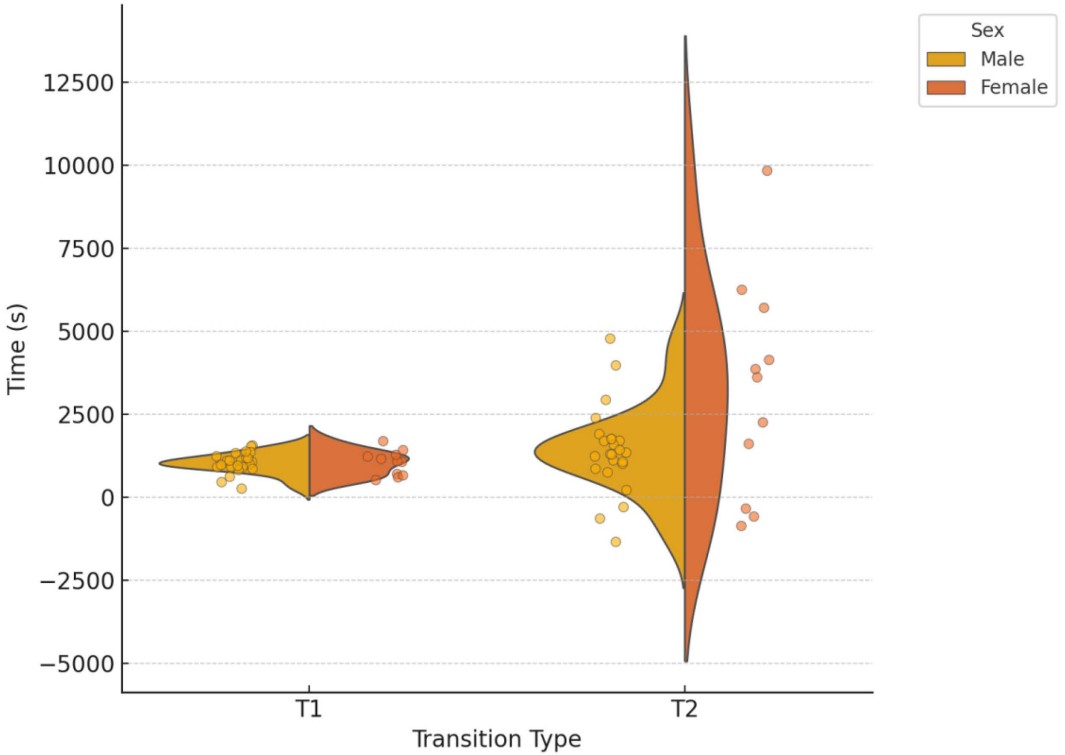

**Fig 2. Distribution of transition times, for Transition 1 (left) and Transition 2 (right).**

The comparisons between the groups indicated significant and large-magnitude differences across various quartile and sex combinations (Table 5). For cycling, the men in Q1 performed better than the women in Q4, with significant differences (t = −8.48, p = 0.008, d = 7.02). Differences were also observed among women in the Q4 and Q2 quartiles (t = 19.33, p = 0.025, d = 19.33), emphasizing the variations within the same sex. The group differences were less pronounced for running. Comparisons between women in Q2 and men in Q1 showed considerable differences (t = 2.68, p = 0.08, d = 2.37), but without statistical significance.

**Analysis of performance variability in cycling and running: sex and quartile comparisons**

The coefficients of variation indicate a greater dispersion of running times (CV = 58.40%) compared to cycling (CV = 40.72%) in the total sample (Table 6). In cycling, men showed a higher variability (CV = 43.44%) than women (CV = 36.13%), while in running the values were similar between sexes (CV = 57.46% for men and CV = 57.11% for women). Among the quartiles, the greatest variability was observed in Q1 for both modalities (CV = 25.26% in cycling and CV = 111.54% in running), indicating a greater dispersion among the best performances. The intermediate quartiles showed less dispersion, particularly in cycling (Q2: CV = 5.20%) and running (Q2: CV = 6.59%).

The individual CVs reveal differences in performance variability between male and female athletes (Table 7). For men, CV values ranged from 0.240 (last man) to 0.434 (first man), indicating a wide range of performance consistency. For women, the CV dispersion was lower, ranging from 0.283 (penultima women) to 0.4045 (third-placed women). These results suggest that although both sexes exhibit variability, men demonstrate a greater range of variation in individual performance.

**Table 3. Descriptive statistics of cycling and running lap times by sex. Times are expressed in seconds (SI).**

| Sample | Total | | Men | | Women | |
|---|---|---|---|---|---|---|
| **Variables** | **Cycling (s)** | **Running (s)** | **Cycling (s)** | **Running (s)** | **Cycling (s)** | **Running (s)** |
| Count | 35 | 35 | 24 | 24 | 11 | 11 |
| Mean | 978.40 | 750.88 | 985.60 | 740.90 | 962.69 | 771.75 |
| SD | 178.36 | 72.09 | 183.79 | 67.19 | 173.42 | 80.69 |
| Minimum | 355.83 | 611.99 | 355.83 | 611.99 | 628.50 | 658.04 |
| 25% | 946.55 | 706.46 | 972.73 | 708.05 | 847.05 | 708.57 |
| 50% | 1034.40 | 737.57 | 1036.70 | 729.11 | 1034.40 | 778.53 |
| 75% | 1091.87 | 815.26 | 1091.65 | 775.68 | 1093.32 | 837.26 |
| Maximum | 1151.29 | 895.36 | 1151.29 | 861.97 | 1130.32 | 895.36 |

**Table 4. Effects of sex and quartiles on cycling and running performance.**

| | sum_sq | F | p | η² |
|---|---|---|---|---|
| C(Sex) | 107,446.58 | 0.0004 | 0.9835 | 0.0 |
| C(Cycling_Quartile) | 26,364,033,777.35 | 35.802 | <0.001 | 0.383 |
| C(Sex):C(Cycling_Quartile) | 26,432,764,855.67 | 35.895 | <0.001 | 0.384 |
| C(Sex) | 6,534,566,764.92 | 8.874 | 0.0005 | 0.095 |
| C(Run_Quartile) | 4,241,088,255.66 | 5.759 | 0.0049 | 0.061 |
| C(Sex):C(Run_Quartile) | 5,154,606,751.86 | nan | nan | 0.075 |

**Table 5. Group comparisons of performance across quartiles and sexes in cycling and running.**

| Metric | Group 1 | Group 2 | t | p | Cohen's d | Interpretation | Absolute Difference (s) | % Difference |
|---|---|---|---|---|---|---|---|---|
| Cycling | Q1_Male | Q2_Female | −2.908 | 0.099 | 2.376 | Large | 147.0 | 0.45 |
| | Q1_Male | Q4_Female | −8.481 | 0.008 | 7.022 | Large | 20669.0 | 93.04 |
| | Q1_Male | Q3_Male | −4.711 | 0.026 | 3.932 | Large | 8031.67 | 28.15 |
| | Q2_Female | Q4_Female | −19.327 | 0.025 | 19.327 | Large | 20522.0 | 92.68 |
| | Q2_Female | Q3_Male | −5.458 | 0.105 | 5.458 | Large | 7884.67 | 27.7 |
| | Q4_Female | Q3_Male | 7.785 | 0.018 | 7.785 | Large | 12637.33 | 69.43 |
| | Q3_Male | Q4_Female | −7.785 | 0.018 | 7.785 | Large | 12637.33 | 69.43 |
| Running | Q4_Male | Q2_Female | 2.617 | 0.227 | 2.617 | Large | 6592.33 | 15.31 |
| | Q4_Male | Q1_Male | 3.054 | 0.195 | 3.043 | Large | 4046.0 | 9.68 |
| | Q2_Female | Q1_Male | 2.679 | 0.080 | 2.372 | Large | 2546.33 | 5.65 |

The Fig 3 illustrates the distribution of the coefficient of variation (CV) percentage in performance among male and female athletes, using a boxplot. The CV represents the relative variability in lap times, with higher values indicating greater inconsistency in performance.

## Discussion

This study aimed to investigate sex differences in performance and pacing at the largest and fastest World Championship in Quintuple Iron ultra-triathlon. The main findings were that (i) men were slower overall in cycling and swimming, and faster in running and overall race time than the women, (ii) the variability in lap times was similar in cycling for both women

**Table 6. Coefficients of variation for cycling and running times by sex and performance quartiles.**

| Metric | Group Type | Group | CV |
|---|---|---|---|
| Cycling_Duration | By Quartile | Q1 | 25.264 |
| | | Q2 | 5.195 |
| | | Q3 | 11.678 |
| | | Q4 | 11.245 |
| By Sex | | Female | 36.133 |
| | | Male | 43.442 |
| | | Total Sample | Total | 40.718 |
| Running_Duration | By Quartile | Q1 | 111.542 |
| | | Q2 | 6.590 |
| | | Q3 | 9.099 |
| | | Q4 | 21.493 |
| By Sex | | Female | 57.110 |
| | | Male | 57.460 |
| | | Total Sample | Total | 58.396 |

**Table 7. Individual coefficients of variation in performance by sex and ordered based on total time.**

| Athlete | Sex | CV | Athlete | Sex | CV |
|---|---|---|---|---|---|
| 1st place | Male | 0.434 | 15th place | Male | 0.371 |
| 1st place | Female | 0.295 | 5th place | Female | 0.351 |
| 2nd place | Female | 0.351 | 6th place | Female | 0.333 |
| 2nd place | Male | 0.330 | 16th place | Male | 0.360 |
| 3rd place | Male | 0.325 | 17th place | Male | 0.369 |
| 4th place | Male | 0.323 | 18th place | Male | 0.268 |
| 5th place | Male | 0.338 | 7th place | Female | 0.337 |
| 3rd place | Female | 0.404 | 19th place | Male | 0.359 |
| 6th place | Male | 0.303 | 20th place | Male | 0.348 |
| 7th place | Male | 0.308 | 8th place | Female | 0.323 |
| 4th place | Female | 0.298 | 21st place | Male | 0.354 |
| 8th place | Male | 0.392 | 22nd place | Male | 0.426 |
| 9th place | Male | 0.366 | 23rd place | Male | 0.431 |
| 10th place | Male | 0.316 | 9th place | Female | 0.357 |
| 11th place | Male | 0.353 | 10th place | Female | 0.282 |
| 12th place | Male | 0.312 | 11th place | Female | 0.331 |
| 13th place | Male | 0.355 | 24th place | Male | 0.239 |
| 14th place | Male | 0.288 | | | |

and men, but higher in running for women, (iii) a significant interaction was found between sex and performance quartiles in cycling but not in running, and (iv) higher variability in performance for men compared to women in cycling, but with similar variability in running for both sexes.

## Performance metrics in triathlon disciplines by sex

A key finding in this study was that men had a higher mean cumulative time in cycling and swimming, whereas women exhibited a higher mean time in running. While the total race time did not show statistically significant differences between the

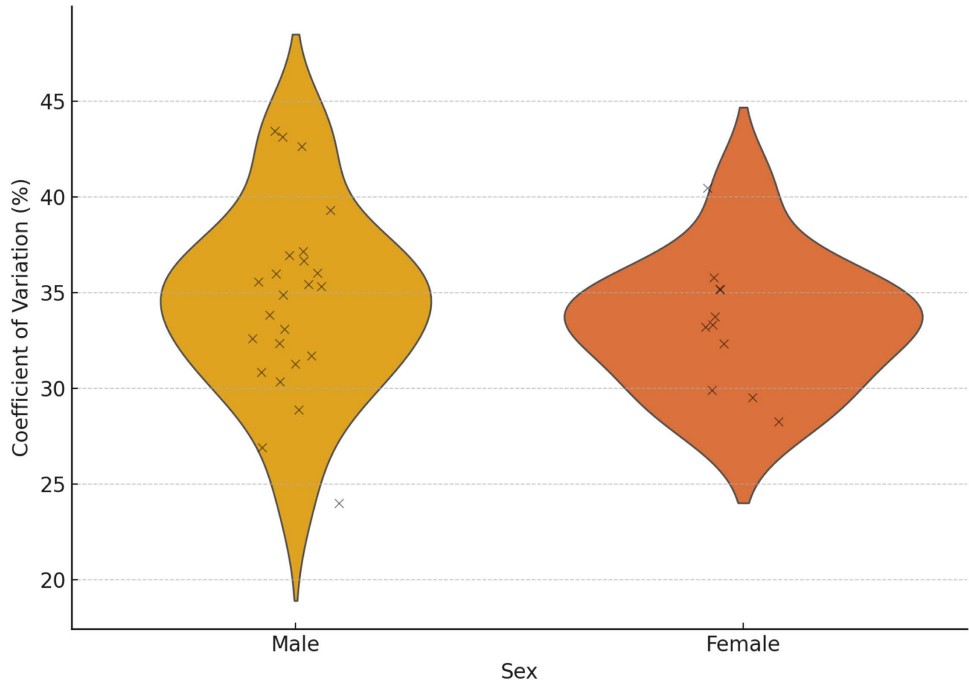

**Fig 3. Boxplot of the Coefficient of Variation (CV) percentage in Performance by Sex.**

sexes, there were notable trends that aligned with or contrasted previous research. In swimming, men were slightly slower than women (Mean: 28,257.5 s vs. 26,770.91 s, respectively) with a small effect size (d = −0.33), suggesting a marginal but consistent difference, which disagreed with previous research on ultra-triathlons showing either no difference [2] or worst swimming performance in women [12]. Particularly, in IRONMAN® triathletes, it has been observed that men are faster than women in swimming independent of age [12]. In ultra-triathlon race covering x-times the IRONMAN® triathlon distance, no significant sex-based differences were found in swimming [2]. However, our findings were in the same line with previous research for single disciplines, where it has been found that sex differences in swimming are minimal or women can even outperform men in long-distance open-water swimming, as women's greater fat stores enhance buoyancy and reduce drag [13].

In cycling, men had a slightly higher mean time (181,113.04 s) compared to women (178,659.18 s), although the effect size was very small (d = −0.08), indicating minimal differences. However, in both the IRONMAN® [12] and the ultra-triathlons [2], men were always faster than women in cycling.

The difference between the sexes was greatest for running, with women achieving a higher mean time (49,789.00 s) than men (45,082.33 s) and a small effect size (d = 0.32) indicating that men were generally faster. This confirms the findings for IRONMAN® [12] and ultra-triathlons of x-times the IRONMAN® distance [2] that men were always faster than women in running.

Running has historically shown greater sex differences than other triathlon disciplines, particularly in ultra-endurance events [14]. While men have a higher absolute speed, women tend to fatigue less over long distances [15]. Studies on ultra-marathon have confirmed that men typically run faster, but women's performance declines at a slower rate, likely due to superior fat oxidation and pacing strategies [16].

The finding that the women were faster than the men in swimming and cycling in this race can be explained by the fact that it was the World Championship with the selection of the best ultra-triathletes in the world and that a very high percentage of women competed and finished the race, again indicating the high level of female athletes [10].

Thus, our findings confirmed that the performance gap in ultra-triathlon has been narrowed, which can be explained by physiological (such as superiority of women in fatigue resistance, substrate efficiency and energetic demands) and other factors [17]. Although it has been recently suggested men presented superior characteristics in terms of speed, muscle strength and power, especially, after puberty [18], it seemed that these characteristics played only minor role in ultra-triathlon. Furthermore, the findings of research on sex differences in performance was subject to the methodological limitation that less women participated in previous studies and races so far [19].

## Lap time performance metrics by sex

We found similar variability in lap times in cycling for both women and men, but higher variability in running for women. From a practical perspective, this means that both women and men maintained a relatively consistent pacing when cycling, while women took more breaks than men when running. The analysis of lap time performance metrics by sex revealed that the variability in cycling lap times was similar for both men and women, whereas women exhibited higher variability in running lap times. Again, this reinforces that while both sexes maintained relatively consistent pacing in cycling, women tended to insert more breaks when running compared to men [20].

The greater variability in women's running lap times may be attributed to differences in fatigue management, pacing strategies or race conditions [21–23]. Prior research has shown that women often adopt more conservative pacing strategies in ultra-endurance events, allowing for brief recovery periods to sustain performance over extended periods of time [23]. In contrast, men often exhibit a more aggressive pacing strategy with fewer breaks but a more significant slowdown later in the race [24,25], which aligns with previous findings in ultra-marathons [26] and IRON-MAN® triathlons [10].

Cycling lap times, which were similarly consistent across sexes, support previous research suggesting that sex differences in pacing are less pronounced in cycling compared to running, as it is a discipline where energy expenditure can be more evenly distributed over long periods of time [2]. The observed greater variability in women's running lap times could be influenced by a combination of muscle fatigue resistance, metabolic efficiency and pacing adjustments in response to environmental factors [27,28]. Previous studies on long-distance triathlons [11] and ultra-marathons [26] have indicated that women tend to slow down more gradually than men, but that they also manage fatigue more effectively through strategic pace adjustments [15], which could explain the higher variability observed in their running lap times. These findings contribute to the broader understanding of sex-based differences in pacing variability, particularly in ultra-triathlon events where endurance management is a key determinant of success.

## Influence of sex and performance quartiles on cycling and running times

The analyses showed a significant interaction between sex and performance quartiles in cycling, but not in running. In other words, cycling demonstrated greater overall discrepancies, whereas running showed more homogeneous differences across quartiles and sexes. These results highlight the greater variability and more evident contrasts in cycling performance between quartiles and sexes. Differences in cycling performance were more strongly influenced by the athlete's overall level (quartile), whereas running performance remained relatively stable across different performance levels for both sexes. These findings are consistent with previous research showing that cycling performance in ultra-endurance events such as ultra-cycling [29,30] or ultra-triathlon [11] is more affected by individual pacing strategies, energy conservation tactics and external conditions such as terrain and drafting effects [5]. The significant interaction effect in cycling suggests that higher-performing cyclists maintained a more even pace. In contrast, lower-performing athletes exhibited greater variability in lap times, possibly due to fatigue management, pacing inefficiencies or race experience. This finding is consistent with research suggesting that elite endurance cyclists/triathletes use more consistent pacing strategies. In contrast, lower-performing cyclists demonstrate greater fluctuations in effort [2,11].

In contrast, running performance was more homogenous across quartiles and sexes, i.e., faster and slower runners followed similar pacing trends, leading to less pronounced sex-based differences. This may be due to the physiological demands of ultra-endurance running, where both men and women must adopt fatigue-resistant strategies over long periods of time [31]. Unlike cycling, where external factors (*e.g.,* wind resistance, terrain, and equipment) can play a significant role in performance variability, running is primarily dictated by internal physiological constraints such as energy depletion, muscular endurance, and hydration levels [32]. Previous studies on ultra-marathons [26] and IRONMAN® triathlon events [10] have shown that women tend to maintain a more stable running pace. In contrast, men demonstrate expressive variations, but these tend to diminish during extreme endurance events [16].

The greater variability in cycling performance quartiles compared to running suggests that cycling is more dependent on training level, experience and pacing strategy, whereas running endurance is more evenly distributed across competitors [33]. This insight contributes to a broader understanding of pacing optimization in ultra-triathlons and highlights the importance of race-specific training approaches for cycling compared to running in endurance sports. Future research could further explore how aerobic capacity, power output and pacing discipline contribute to performance variability across quartiles in ultra-endurance triathlons.

## Analysis of performance variability in cycling and running: sex and quartile comparisons

A final key finding was a higher variability in cycling performance for men compared to women, with similar variability in running for both sexes. Men also demonstrated a greater range of variation in individual performance than women in overall race times. From a practical perspective, the 11 women showed a higher density in performance compared to the 24 men, i.e., the female sample was more highly selected than the male sample. These findings are consistent with previous research indicating that men tend to adopt more aggressive pacing strategies in endurance cycling events, leading to greater variability in performance. At the same time, women often maintain a steadier and more controlled pace [34].

A significant observation was that the overall race time distribution for men was broader than for women, meaning that some male athletes performed exceptionally well. Others, on the other hand, experienced significant declines in performance. This greater variability in men's performance could be attributed to differences in race experience, pacing discipline and physiological resilience. Research on ultra-endurance sports suggests that women are more likely to regulate their pace effectively, reducing unnecessary fluctuations in effort, while men tend to experience greater drops in performance due to early overexertion [35]. This pattern is evident in IRONMAN® triathlons, where women's pacing tends to be more even, while men show greater fluctuations in pace across different disciplines [10].

From a practical standpoint, the higher density of female performance suggests a more selective group of participants. With only 11 female finishers compared to 24 men, it is likely that the women who completed the race were among the more experienced and better-prepared competitors, whereas the larger number of male finishers allowed for a wider range of abilities and pacing strategies, resulting in greater performance variability. This aligns with findings from long-distance running events, where female competitors tend to have a higher baseline endurance level due to race selection effects, meaning that only the best well-prepared women tend to participate and complete ultra-endurance events [31].

Similar variability in running performance between the sexes suggests that running performance in ultra-triathlons is determined by physiological endurance constraints rather than pacing differences [17]. Studies have shown that women may experience less fatigue-induced performance declines in ultra-endurance running, allowing them to maintain a more consistent pace over a longer period of time [36,37]. This could explain why running variability did not differ significantly between men and women, as both sexes may have adapted their strategies to maximize energy conservation and minimize declines in speed over time. Although pacing decisions may partly explain this, the smaller female sample may have amplified visual spread in data

## Limitations

Although we were able to analyze a very large data set of ultra-triathletes, this study is not free of limitations. Aspects that influence ultra-endurance performance, such as previous experience [27], pre-race preparation [38], training [39], nutrition [40], sleep management [41] and support [42] during the race, could not be considered. The manual timing in swimming introduces a potential source of human error not present in electronically timed disciplines. On the other hand, strength of the study was its novelty as it provided insights on the sex difference in sport performance using the paradigm of a very demanding ultra-triathlon. Unlike cycling and running, swimming times were recorded manually, which may introduce measurement variability. Additionally, the unequal sample sizes between men (n = 24) and women (n = 11) may influence the statistical power of sex-based comparisons and should be considered when interpreting the results. Considering the increasing popularity of ultra-triathlon and the entry of more women in this race, the findings would have practical applications on providing guidance of women athletes. Coaches and athletes can refine training plans and pacing strategies based on discipline- and sex-specific trends, and sport scientists will find this dataset a valuable benchmark for modeling human endurance at the limits.

## Conclusion

In summary, this analysis of the men and women who competed in the fastest and largest World Championship in Quintuple Iron ultra-triathlon to date shows that men were slower in swimming and cycling but faster in running and overall race time. Female outperformance in this specific event, which is likely due to the higher performance density among female competitors, is reflected in swimming and cycling. Lap time variability was similar for both women and men in cycling but higher for women in running, with a significant interaction between sex and performance quartiles in cycling but not in running. These differences may be explained by variations in race tactics and/or physiological differences between women and men. The higher performance variability in men compared to women in cycling, alongside similar variability in running for both sexes, could stem from differing race tactics in these two disciplines. The findings suggest that women's endurance strategies may contribute to their competitive edge in ultra-triathlons, emphasizing the need for sex-specific training and pacing approaches for optimal performance in ultra-endurance events.

## Supporting information

**S1 File. Data.**
(XLSX)

## Author contributions

**Conceptualization:** Beat Knechtle.

**Data curation:** Beat Knechtle.

**Formal analysis:** Luciano Bernardes Leite, Pedro Forte.

**Writing – original draft:** Beat Knechtle, Pedro Forte.

**Writing – review & editing:** Luciano Bernardes Leite, Sasa Duric, Ivan Cuk, Marilia S. Andrade, Volker Scheer, Pantelis T. Nikolaidis, Katja Weiss, Thomas Rosemann.

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
