## [Decision Letter · Decision Letter 0]

15 Jul 2025

Dear Dr. Knechtle,

Thank you for submitting your manuscript to PLOS ONE. After careful consideration, we feel that it has merit but does not fully meet PLOS ONE’s publication criteria as it currently stands. Therefore, we invite you to submit a revised version of the manuscript that addresses the points raised during the review process.

We look forward to receiving your revised manuscript.

Kind regards,

Ratko Peric, PhD

Academic Editor

PLOS ONE

Journal Requirements:

Reviewers' comments:

Reviewer's Responses to Questions

**Comments to the Author**

1. Is the manuscript technically sound, and do the data support the conclusions?

Reviewer #1: Yes

Reviewer #2: Yes

2. Has the statistical analysis been performed appropriately and rigorously?

Reviewer #1: Yes

Reviewer #2: Yes

3. Have the authors made all data underlying the findings in their manuscript fully available?

Reviewer #1: Yes

Reviewer #2: Yes

4. Is the manuscript presented in an intelligible fashion and written in standard English?

Reviewer #1: Yes

Reviewer #2: Yes

Reviewer #1: General Assessment

This manuscript tackles an ambitious and timely question in ultra-endurance sport by analyzing the largest and fastest Quintuple Iron ultra-triathlon ever held, with 35 finishers (11 women, 24 men). The authors have assembled an impressive dataset to explore how pacing varies across the disciplines (swim, cycling, run) and between sexes. Their findings—especially the record‐breaking female performances—underscore the value of such large‐scale analyses. Although the study has certain limitations (see Limitations section), the insights provided will be of high practical value to athletes, coaches, and exercise scientists investigating human performance in extreme sports.

Major Recommendations

1) Enhance Data Presentation in Figures 1 & 2

Include Raw Data Points: To adhere to the more informative presentation paradigm (Weissgerber et al. 2015), overlay individual data points or violin/box‐and‐scatter combinations behind the bar/line summaries.

Correct Y-Axis Units: Ensure the Y-axis is labeled with the precise unit of time used throughout (e.g., “Time (s)” , as appropriate).

2) p-Values in Table 4

Replace any “0.0” entries in the p-value column with an inequality (e.g., “<0.001”) to avoid implying absolute zero and to conform to standard statistical reporting.

3) Streamline and Augment Table 5

Remove Redundant Comparisons: Omit trivial intra-group comparisons (e.g., “Q1_Male vs. Q1_Male”).

Add Effect Sizes: For each relevant pairwise comparison, include columns for the absolute difference (in seconds) and the relative difference (percentage), which will greatly aid practical interpretation.

4) Figure 4 Adjustments

Add “(%)” to the Y-axis label for clarity. Apply the same data-presentation rules as for Figures 1 and 2 (raw data points plus summary statistics).

Minor Corrections

Line 81 : Change the stated run distance from “41.2 km” to the correct “42.2 km.”

Practical Implications

Although the authors candidly discuss methodological constraints (e.g., potential selection bias, incomplete split data), their analyses offer actionable guidance. Coaches and athletes can refine training plans and pacing strategies based on discipline- and sex-specific trends, and sport scientists will find this dataset a valuable benchmark for modeling human endurance at the limits.

Summary

With the above revisions—particularly improving transparency in figures and tables—the manuscript will achieve a higher level of scientific rigor and practical utility. I look forward to seeing these enhancements in the revised submission.

Reviewer #2: Manuscript Number: PONE-D-25-29597

Manuscript Title: Sex differences in performance and pacing in the greatest Quintuple Iron ultra-triathlon race in history – the IUTA World Championship 2024 in France

The manuscript is interesting and well written, and well-structured, with several strengths. It represents one of the few studies to utilize detailed data on pacing strategies among ultra-triathletes competing in the fastest and largest Quintuple Iron Ultra-Triathlon World Championship ever held. The study offers novel and valuable insights into sex-based performance dynamics in ultra-endurance sport. The findings suggest that female athletes may benefit from specific endurance strategies, potentially contributing to their competitive advantage in such extreme events. This highlights the importance of developing sex-specific training and pacing approaches to optimize performance in ultra-endurance competitions.

Overall, the different sections of the manuscript are well written. I just have a few minor comments to make.

- The unequal sample sizes between men (n=24) and women (n=11) may influence the statistical power ?. I recommend that the authors explicitly mention this in the limitations section.

- Please use consistent terminology throughout the manuscript. Although "sex differences" and "gender differences" are sometimes used interchangeably, the term "sex" is preferred in scientific sports physiology when referring to biological distinctions.

- Line 57: I suggest adding the characteristics of the participants (e.g., age).

- Lines 67 – 37: This paragraph is too long, consider to create small paragraphs.

- Lines 339 – 340: change “there is a significant interaction was found between sex and performance quartiles in cycling but not in running” to “a significant interaction was found between sex and performance quartiles in cycling but not in running”.

**Do you want your identity to be public for this peer review?** For information about this choice, including consent withdrawal, please see our Privacy Policy

Reviewer #1: No

Reviewer #2: **Yes**

---

## [Author Response · Author response to Decision Letter 1]

18 Jul 2025

Reviewer #1: General Assessment

This manuscript tackles an ambitious and timely question in ultra-endurance sport by analyzing the largest and fastest Quintuple Iron ultra-triathlon ever held, with 35 finishers (11 women, 24 men). The authors have assembled an impressive dataset to explore how pacing varies across the disciplines (swim, cycling, run) and between sexes. Their findings—especially the record‐breaking female performances—underscore the value of such large‐scale analyses. Although the study has certain limitations (see Limitations section), the insights provided will be of high practical value to athletes, coaches, and exercise scientists investigating human performance in extreme sports.

Major Recommendations

1) Enhance Data Presentation in Figures 1 & 2

Include Raw Data Points: To adhere to the more informative presentation paradigm (Weissgerber et al. 2015), overlay individual data points or violin/box‐and‐scatter combinations behind the bar/line summaries.

Correct Y-Axis Units: Ensure the Y-axis is labeled with the precise unit of time used throughout (e.g., “Time (s)” , as appropriate).

Answer: We agree with the expert reviewer and we included violin with scatter combinations.

2) p-Values in Table 4

Replace any “0.0” entries in the p-value column with an inequality (e.g., “<0.001”) to avoid implying absolute zero and to conform to standard statistical reporting.

Answer: We agree with the expert reviewer, and we have corrected it.

3) Streamline and Augment Table 5

Remove Redundant Comparisons: Omit trivial intra-group comparisons (e.g., “Q1_Male vs. Q1_Male”).

Add Effect Sizes: For each relevant pairwise comparison, include columns for the absolute difference (in seconds) and the relative difference (percentage), which will greatly aid practical interpretation.

Answer: We agree with the expert reviewer and corrected reducing the redundances and adding the absolute and relative differences.

4) Figure 4 Adjustments

Add “(%)” to the Y-axis label for clarity. Apply the same data-presentation rules as for Figures 1 and 2 (raw data points plus summary statistics).

Answer: We believe that the reviewer meant figure 3. We agree with the expert reviewer and we changed figure 3 in agreement with figure 1 and 2.

Minor Corrections

Line 81: Change the stated run distance from “41.2 km” to the correct “42.2 km.”

Answer: We agree with the expert reviewer and corrected as suggested.

Practical Implications

Although the authors candidly discuss methodological constraints (e.g., potential selection bias, incomplete split data), their analyses offer actionable guidance. Coaches and athletes can refine training plans and pacing strategies based on discipline- and sex-specific trends, and sport scientists will find this dataset a valuable benchmark for modeling human endurance at the limits.

Answer: We agree with the expert reviewer and included your suggestions in practical applications

Summary

With the above revisions—particularly improving transparency in figures and tables—the manuscript will achieve a higher level of scientific rigor and practical utility. I look forward to seeing these enhancements in the revised submission.

Answer: We thank the expert reviewer for the comments and hope that our corrections are appropriate.

Reviewer #2: Manuscript Number: PONE-D-25-29597

Manuscript Title: Sex differences in performance and pacing in the greatest Quintuple Iron ultra-triathlon race in history – the IUTA World Championship 2024 in France

The manuscript is interesting and well written, and well-structured, with several strengths. It represents one of the few studies to utilize detailed data on pacing strategies among ultra-triathletes competing in the fastest and largest Quintuple Iron Ultra-Triathlon World Championship ever held. The study offers novel and valuable insights into sex-based performance dynamics in ultra-endurance sport. The findings suggest that female athletes may benefit from specific endurance strategies, potentially contributing to their competitive advantage in such extreme events. This highlights the importance of developing sex-specific training and pacing approaches to optimize performance in ultra-endurance competitions.

Overall, the different sections of the manuscript are well written. I just have a few minor comments to make.

- The unequal sample sizes between men (n=24) and women (n=11) may influence the statistical power ?. I recommend that the authors explicitly mention this in the limitations section.

Answer: We agree with the expert reviewer and have now explicitly acknowledged in the limitations section that the unequal sample sizes between men (n = 24) and women (n = 11) may influence the statistical power of the comparisons performed. We added ‘Additionally, the unequal sample sizes between men (n = 24) and women (n = 11) may influence the statistical power of sex-based comparisons and should be considered when interpreting the results’.

- Please use consistent terminology throughout the manuscript. Although "sex differences" and "gender differences" are sometimes used interchangeably, the term "sex" is preferred in scientific sports physiology when referring to biological distinctions.

Answer: We agree with the expert reviewer and used sex. The term gender is now replaced throughout the manuscript.

- Line 57: I suggest adding the characteristics of the participants (e.g., age).

Answer: We agree with the expert reviewer and included the info.

- Lines 67 – 37: This paragraph is too long, consider to create small paragraphs.

Answer: We agree with the expert reviewer and we summarized the paragraph.

- Lines 339 – 340: change “there is a significant interaction was found between sex and performance quartiles in cycling but not in running” to “a significant interaction was found between sex and performance quartiles in cycling but not in running”.

Answer: We agree with the expert reviewer and corrected as suggested.

---

## [Decision Letter · Decision Letter 1]

19 Aug 2025

Sex differences in performance and pacing in the greatest Quintuple Iron ultra-triathlon race in history – the IUTA World Championship 2024 in France

PONE-D-25-29597R1

Dear Dr. Knechtle,

We’re pleased to inform you that your manuscript has been judged scientifically suitable for publication and will be formally accepted for publication once it meets all outstanding technical requirements.

Kind regards,

Ratko Peric, PhD

Academic Editor

PLOS ONE

Additional Editor Comments (optional):

Reviewers' comments:

Reviewer's Responses to Questions

**Comments to the Author**

Reviewer #1: All comments have been addressed

Reviewer #2: All comments have been addressed

2. Is the manuscript technically sound, and do the data support the conclusions?

Reviewer #1: Yes

Reviewer #2: Yes

3. Has the statistical analysis been performed appropriately and rigorously?

Reviewer #1: Yes

Reviewer #2: Yes

4. Have the authors made all data underlying the findings in their manuscript fully available?

Reviewer #1: Yes

Reviewer #2: Yes

5. Is the manuscript presented in an intelligible fashion and written in standard English?

Reviewer #1: Yes

Reviewer #2: Yes

Reviewer #1: Thank you for carefully addressing all of the reviewer’s comments in the revised manuscript. The changes are clear, appropriate, and have strengthened the paper’s clarity and rigor.

Congratulations on an interesting and valuable piece of work. I wish you every success with publication and the continuation of this research.

Reviewer #2: The manuscript is highly interesting and nicely written. The authors have been well corrected and modified the manuscript according to my comments.

I recommend to accept the manuscript for publication

**Do you want your identity to be public for this peer review?** For information about this choice, including consent withdrawal, please see our Privacy Policy

Reviewer #1: No

Reviewer #2: **Yes**

---

## [Editor Report · Acceptance letter]

PONE-D-25-29597R1

PLOS ONE

Dear Dr. Knechtle,

I'm pleased to inform you that your manuscript has been deemed suitable for publication in PLOS ONE. Congratulations! Your manuscript is now being handed over to our production team.

Kind regards,

on behalf of

Dr. Ratko Peric

Academic Editor

PLOS ONE